# Disparities in chronic kidney disease burden estimates: From different sources, definitions, and equations

Yao Ma[1], Xiang Wang[2], Weihong Zhao ●[1]*

1 Division of Nephrology, Department of Geriatrics, Jiangsu Province Hospital and Nanjing Medical University First Affiliated Hospital, Nanjing, China, 2 Department of Biostatistics, School of Public Health, Nanjing Medical University, Nanjing, China

* zhaoweihongny@njmu.edu.cn

## Abstract

### Introduction

The Global Burden of Disease (GBD) study provides updated epidemiological descriptions of chronic kidney disease (CKD), yet the discrepancies between its estimates and those from other sources remain unclear. Furthermore, attention is required due to the specificity of standard and computational tool for glomerular filtration rate (GFR). We aimed to evaluate CKD burden from various sources, definitions, and equations.

### Methods

This study analyzed CKD prevalence among US adults from 1999 to 2018, using data from the GBD study 2021 and the National Health and Nutrition Examination Survey (NHANES). We calculated average prevalence and estimated annual percentage change (EAPC) for the total population and by sex. The analysis was repeated using different definitions and equations. Additionally, a literature review was conducted to extend the comparison to a global scale.

### Results

Among US adults, the annual average estimates from the GBD and NHANES were similar, while long-term trends diverged, with disparities becoming more evident in sex-specific subgroups. Removal of racial coefficients led to an increase in the estimated CKD prevalence in Black individuals, while a decrease was observed in White individuals. The EKFC equation yielded the highest average and single-cycle CKD prevalence. Applying age-adapted thresholds reduced the prevalence of low estimated GFR (eGFR<threshold(s)) by approximately 50%, with numbers of older women reclassified into non-CKD categories.

**Data availability statement:** All data are available on the Global Health Data Exchange (GHDx) platform (https://ghdx.healthdata.org/) and the centers for Disease Control and Prevention (CDC) website (https://wwwn.cdc.gov/nchs/nhanes).

**Funding:** This study was supported by National Key Research and Development Program of China (2023YFC3605500) and National Natural Science Foundation of China (82171585, 81971320). The funders had no role in study design, data collection and analysis, decision to publish, or preparation of the manuscript.

**Competing interests:** The authors declare that they have no competing interests.

## Conclusions

This study highlights the differences in estimated CKD prevalence across various sources. Age-adjusted thresholds and the flexible EKFC equation hold promise for future applications in both epidemiological research and clinical practice.

## Introduction

Chronic kidney disease (CKD) constitutes a significant and escalating public health concern, closely linked to the increased risk of cardiovascular events, end-stage renal disease, and premature mortality [1]. Data from the International Society of Nephrology Global Kidney Health Atlas (ISN-GKHA) indicates that the global median prevalence of CKD stands at 9.5% (IQR 5.9–11.7) [2]. Given its significant adverse effects on prognosis, quality of life, and healthcare resources, appropriate monitoring and management of CKD cannot be overstated.

The Global Burden of Disease (GBD) study provides regularly updated epidemiological estimates of infectious and non-communicable diseases, as well as associated risk factors, playing a pivotal role in academic research and policy development [3,4]. With the release of GBD study 2021 and the upcoming 2023 version, interest in and utilization of this database have reached unprecedented levels. However, it is crucial to acknowledge, though often overlooked, that the database has inherent limitations. Recent two comparative studies between GBD and real-world data have revealed that the GBD framework tends to significantly overestimate the burden of acute infectious diseases and fails to capture temporal fluctuations [5,6]. While the systematic analytical framework incorporating smoothing techniques makes the database theoretically more suitable for chronic diseases, it has seldom been compared with results from other sources. Furthermore, concerning CKD, attention is required due to the specificity of standard (single or age-adapted thresholds) and computational tool (various equations) for estimated glomerular filtration rate (eGFR) [7].

This study aimed to compare CKD burden estimates derived from different databases, which also include comparisons with data from other sources through a comprehensive literature review. Additionally, we assessed the potential impact of using different equations and definitions, with the goal of providing valuable insights for the advancement of epidemiological research and public health strategies.

## Materials and methods

### Study population

This study primarily focused on the prevalence of CKD within US adults, utilizing data from the GBD study 2021 and the National Health and Nutrition Examination Survey (NHANES). To mitigate the potential impact of COVID-19, a 20-year period from 1999 to 2018 was selected for analysis.

Estimates of the annual burden of CKD and its 95% uncertainty intervals (UIs) from the GBD study were obtained from the Global Health Data Exchange, coordinated by the Institute for Health Metrics and Evaluation at the University of

Washington (https://vizhub.healthdata.org/gbd-results/). The GBD study uses deidentified data, with a waiver of informed consent approved by the University of Washington Institutional Review Board. The methods for estimating CKD prevalence have been described in detail [8]. Briefly, Bayesian regression models were applied to data sourced from vital registration systems, end-stage renal disease registries, household surveys, and published literature. The overall framework and search strategy can be found in the previous publication [9]. The 2021 update re-extracted data from the European Renal Association–European Dialysis and Transplant Association (ERA-EDTA) covering 1998–2017, incorporating a global sex coefficient, more detailed dialysis staging, and narrower age ranges, all of which improved estimation accuracy [10].

The NHANES is an ongoing national cross-sectional survey that collects health-related information from US adults and children every two years. Participants are randomly selected through a complex, multistage, cluster-sampling probability design. They are initially interviewed at home and then invited for various examinations and to provide blood samples. Sampling weights account for oversampling, non-coverage, and non-response in specific populations, allowing extrapolation (weighting) to provide national estimates representative of the entire US population. The National Centers for Health Statistics Ethics Review Board approved the study protocol and each participant provided written informed consent. These data are available on the Centers for Disease Control and Prevention (CDC) website (https://wwwn.cdc.gov/nchs/nhanes).

## Definitions

Adult CKD is diagnosed based on persistent kidney damage, including elevated urinary albumin-to-creatinine ratio (ACR) and/or a GFR below the specified threshold. In the GBD study, CKD was defined as a single estimate of GFR < 60 ml/min/1.73m² or ACR > 30 mg/g, which encompasses individuals with end-stage renal disease who are undergoing dialysis or have received a transplant. However, GFR varies with age. In healthy populations, 40 years of age marks a critical threshold, with no age-dependent decline in renal function observed prior to this point [11]. Therefore, the single threshold of 60 ml/min/1.73m² is limited and controversial, as it often leads to underidentification of pathology in younger adults and overdiagnosis of CKD in older individuals [12]. Therefore, in NHANES, we applied an age-adapted definition, developed based on age-specific percentiles of estimated or measured GFR or age-adjusted thresholds, specifically: 75 ml/min/1.73m² for individuals under 40 years, 60 ml/min/1.73m² for those between 40 and 64 years, and 45 ml/min/1.73m² for individuals above 65 years.

## GFR estimating equations

Due to their complexity and cost, the gold-standard method (inulin urinary clearance) and other reference methods (including iohexol, iothalamate, and isotopes) are difficult to implement at scale. As a result, GFR estimating equations are often the preferred approach for assessing renal function [13]. In the GBD study 2021, the creatinine-based Chronic Kidney Disease Epidemiology Collaboration (CKD-EPI) equation developed in 2009 (CKD-EPI$_{2009}$) was designated as the reference equation for estimating adult GFR [14], with eGFR data from other equations adjusted accordingly. Given that race is a complex social construct and to reduce disparities, the CKD-EPI$_{2021}$ equation eliminates the race coefficient [15]. This updated equation is recommended for use by the American Society of Nephrology, the National Kidney Foundation, and the American Association for Clinical Chemistry. Additionally, we included the newly developed European Kidney Function Consortium (EKFC) equation [16], which has recently been validated in the US population using US Q-values [17]. Detailed expressions of these equations are provided in S1 Table.

## Statistical analysis

In the GBD study 2021, we extracted data on CKD cases and prevalence for US adults aged 20 and older from 1999 to 2018. We calculated the 20-year and 10-year average prevalence for the total population and sex subgroups. The estimated annual percentage change (EAPC) was determined by fitting a regression line to the natural logarithm of

the annual prevalence, expressed as ln(y) = α + βx + ε, where 'y' represents the annual prevalence rate and 'x' denotes the calendar year. The EAPC was then calculated using the formula 100 × (e^β − 1). In NHANES, we extracted basic demographic information, serum creatinine levels, urine albumin levels, and urine creatinine levels from all participants (1999–2018, 10 cycles). Individuals with missing data or those under 20 years of age were excluded (detailed inclusion and exclusion process was shown in S1 Fig). Serum creatinine levels were adjusted according to the recommendations provided in NHANES (S2 Table). We accounted for the complex sampling design and weights, and repeated the above calculations using different definitions and equations (Table 1). All analyses were performed using R 4.4.1. We conducted a literature review to compare the GBD estimates with studies that included nationally representative samples, extending the comparison to a global scale. Details are provided in the Supplementary Materials (S1 File).

## Results

### Differences across databases

Annual estimated CKD cases and prevalence are detailed in S3 Table. GBD estimates for the total population exhibited a consistent upward trend, with the highest prevalence recorded in 2018 at 15.2%. In contrast, CKD burden estimates from NHANES fluctuated, with the lowest prevalence observed in the 2009–2010 survey (12.7%) and the highest in the 2013–2014 survey (15.5%) (Fig 1A). Similar trends were observed within sex subgroups, with a higher prevalence in women. Estimates from the two databases for men peaked in 2018, reaching 12.9% and 14.1%, respectively. For women, the GBD estimates were highest in 2018 at 17.3%, while the NHANES estimates peaked in 2013–2014 at 17.5%. The 20-year and 10-year average prevalence, as well as single-cycle estimates, were comparable across both databases, however, disparities became more pronounced in sex-specific subgroups (Table 2). The fluctuations were also reflected in the EAPC analysis. Although both databases show an upward trend in prevalence, the trend in NHANES was statistically insignificant.

**Table 1. Descriptions of CKD criteria and GFR estimating equations in the KDIGO guidelines, GBD study, and NHANES.**

| | KDIGO 2024 guidelines | GBD study | NHANES |
|---|---|---|---|
| **CKD criteria** | Abnormalities of kidney structure or function, present for a minimum of 3 months (at least one of the following)<br>• Albuminuria (ACR ≥ 30 mg/g)<br>• Urine sediment abnormalities<br>• Persistent hematuria<br>• Electrolyte and other abnormalities due to tubular disorders<br>• Abnormalities detected by histology<br>• Structural abnormalities detected by imaging<br>• History of kidney transplantation<br>• GFR < 60 ml/min/1.73m² | At least one of the following:<br>• eGFR < 60 ml/min/1.73m² (single measurement)<br>• ACR > 30 mg/g (single measurement)<br>• With end-stage renal disease (history of dialysis or kidney transplantation) | At least one of the following:<br>• eGFR < 60 ml/min/1.73m² (single measurement)<br>• ACR > 30 mg/g (single measurement)<br>Age-adapted thresholds for GFR:<br>• < 40 years: 75 ml/min/1.73m²<br>• 40–64 years: 60 ml/min/1.73m²<br>• ≥ 65 years: 45 ml/min/1.73m² |
| **GFR estimating equations** | • It is recommended to use validated GFR estimation equations.<br>• Use of race in the computation of eGFR should be avoided. | • The creatinine-based CKD-EPI equation developed in 2009 (CKD-EPI$_{2009}$) was designated as the reference equation for estimating adult GFR.<br>• eGFR data reported using other equations were adjusted accordingly by a fixed ratio. | • In this study, we employed the CKD-EPI$_{2009}$, CKD-EPI$_{2021}$, and EKFC equations, all of which have been extensively validated. Notably, the CKD-EPI$_{2021}$ and EKFC equations do not incorporate race as a factor.<br>• The EKFC equation is based on rescaled creatinine (Q-value), which represents the median serum creatinine level in a healthy population of any age, sex, or race. As long as the Q-value is available, the EKFC equation is expected to be applicable across diverse populations. |

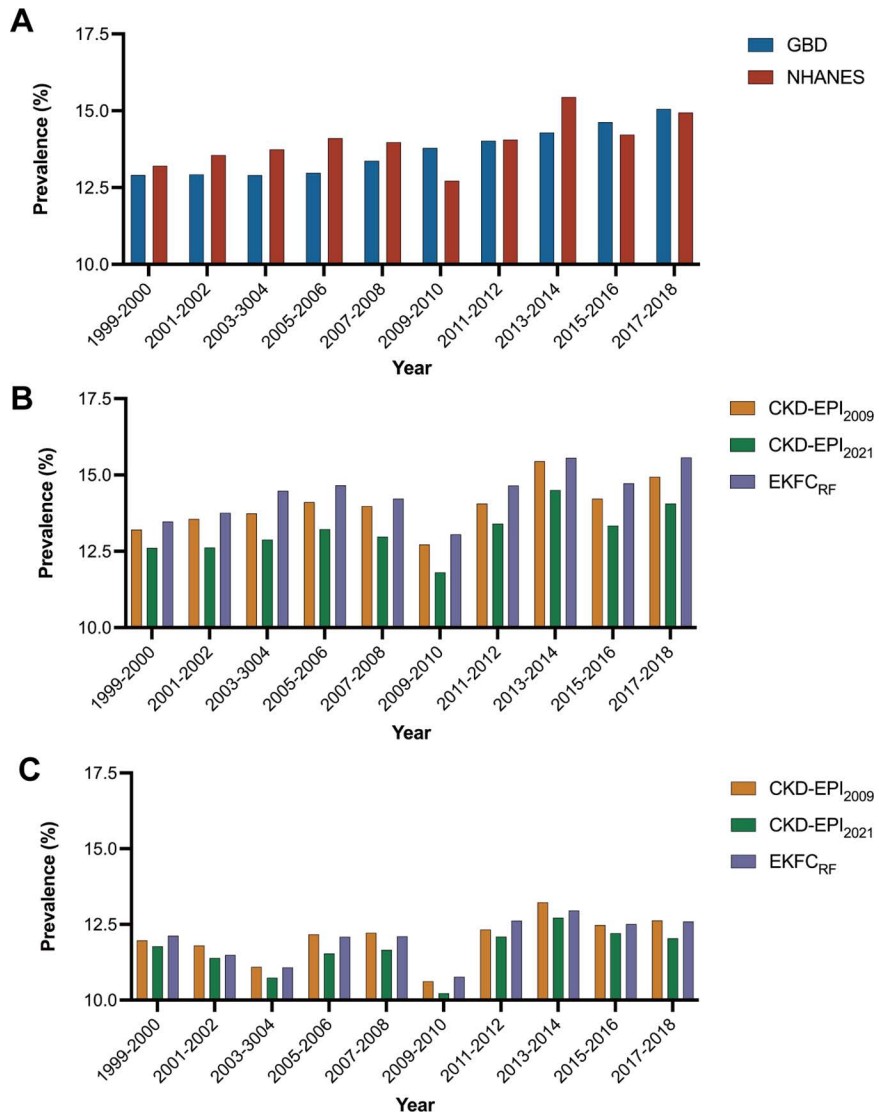

**Fig 1. Comparison of CKD prevalence estimates using different databases, equations, and definitions.** (A) GBD vs. NHANES (using the CKD-EPI_{2009} equation and fixed threshold). (B) CKD-EPI_{2009} vs. CKD-EPI_{2021} vs. EKFC (using the NHANES database and fixed threshold). (c) CKD-EPI_{2009} vs. CKD-EPI_{2021} vs. EKFC (using the NHANES database and age-adapted thresholds).

### Differences across estimating equations

Changes in the GFR estimating equations significantly influenced the estimated prevalence of CKD and low eGFR (eGFR < threshold(s)) (Table 2, Fig 1B, and Table 3). The results obtained using the CKD-EPI_{2021} equation were lower than those from the other two, both for men and women (Table 2). After the removal of race coefficients, the estimated prevalence of low eGFR increased from 6.8% to 9.7% among self-reported Black individuals, while it decreased from 8.5% to 6.4% among self-reported White individuals (Table 3). The EKFC equation yielded the highest average and single-cycle CKD prevalence, primarily due to elevated estimates among women. The prevalence of low eGFR among self-reported Black and White individuals was 11.3% and 9.0%. Among age-stratified subgroups, CKD-EPI_{2021}, CKD-EPI_{2009}, and EKFC equations resulted in the highest prevalence in young, middle-aged, and older adults, respectively.

**Table 2. Annual averages and EAPC of CKD prevalence in US adults using different databases, definitions, and equations.**

| | Fixed threshold | | | | Age-adapted thresholds | | |
|---|---|---|---|---|---|---|---|
| | GBD | NHANES | | | NHANES | | |
| | CKD-EPI$_{2009}$ | CKD-EPI$_{2009}$ | CKD-EPI$_{2021}$ | EKFC$_{RF}$ | CKD-EPI$_{2009}$ | CKD-EPI$_{2021}$ | EKFC$_{RF}$ |
| **1999-2018** | | | | | | | |
| Annual average | 13.7% | 14.0% | 13.1% | 14.4% | 12.1% | 11.6% | 12.0% |
| Males | 11.5% | 12.4% | 11.6% | 12.2% | 10.9% | 10.6% | 10.4% |
| Females | 15.8% | 15.4% | 14.6% | 16.5% | 13.1% | 12.5% | 13.5% |
| EAPC | 0.92 (0.80, 1.04) | 1.17 (−0.01, 2.36) | 1.16 (−0.09, 2.43) | 1.25 (0.08, 2.44) | 1.01 (−0.50, 2.53) | 0.94 (−0.60, 2.50) | 1.06 (−0.33, 2.46) |
| **2009-2018** | | | | | | | |
| Annual average | 14.4% | 14.3% | 13.4% | 14.7% | 12.3% | 11.9% | 12.3% |
| Males | 12.1% | 12.9% | 12.0% | 12.6% | 11.1% | 10.8% | 10.7% |
| Females | 16.4% | 15.6% | 14.8% | 16.7% | 13.3% | 12.7% | 13.8% |
| EAPC | 1.10 (0.98, 1.21) | 3.39 (−2.66, 9.81) | 3.50 (−3.05, 10.49) | 3.65 (−1.56, 9.13) | 3.65 (−3.41, 11.23) | 3.41 (−4.14, 11.54) | 3.10 (−3.40, 10.03) |
| **2017-2018** | | | | | | | |
| Annual average | 15.1% | 14.9% | 14.1% | 15.6% | 12.6% | 12.0% | 12.6% |
| Males | 12.8% | 14.1% | 13.0% | 13.7% | 12.3% | 11.8% | 11.7% |
| Females | 17.2% | 15.8% | 15.0% | 17.3% | 13.0% | 12.0% | 13.4% |

## Differences across definitions

Age-adapted GFR thresholds resulted in a reduction in both the mean and single-cycle CKD prevalence by approximately 10%−20% (Table 2, Fig 1C, and S3 Table). In the 2017–2018 NHANES survey, the prevalence of low eGFR decreased by roughly 50%, and this reduction was more pronounced, reaching 70%, in individuals aged 65 and older (Table 3). Notably, sex differences varied depending on the equation used. When applying a fixed threshold, all three equations indicated a higher prevalence in women. However, with the use of age-adapted GFR thresholds, the CKD prevalence for women was lower than that for men under both CKD-EPI equations. This discrepancy was primarily due to more young men being classified as having CKD and a greater number of older women being reclassified as non-CKD (Table 4).

## Discussion

In this study, we compared CKD burden estimates derived from different databases and found that, among US adults, the annual average estimates from the GBD and NHANES were similar, while long-term trends diverged, with disparities becoming more evident in sex-specific subgroups. Additionally, we assessed the potential impact of using different equations and definitions. Removing the racial coefficient resulted in an increased CKD prevalence estimate for Black individuals, while the prevalence for White individuals decreased. The EKFC equation yielded the highest average and single-cycle CKD prevalence. Applying age-adapted thresholds reduced the prevalence of low eGFR by approximately 50%.

Accurate estimates of the spatiotemporal patterns and trends in CKD prevalence are crucial for understanding the disease burden and improving CKD prevention and management. This study is the first to reveal the disparities in estimated CKD prevalence between the GBD study and other databases or studies that include nationally representative samples. Among US adults, the single-cycle and annual average estimates from the GBD and NHANES were similar, while long-term trends differed. Specifically, in GBD study, the CKD prevalence exhibited a steady annual increase with

**Table 3. The estimated prevalence of low eGFR (eGFR<threshold(s)) among US adults in NHANES (2017-2018).**

| | Fixed threshold | | | Age-adapted thresholds | | |
|---|---|---|---|---|---|---|
| | CKD-EPI$_{2009}$ | CKD-EPI$_{2021}$ | EKFC$_{RF}$ | CKD-EPI$_{2009}$ | CKD-EPI$_{2021}$ | EKFC$_{RF}$ |
| **Total** | 6.91 (5.51, 8.32) | 5.72 (4.73, 6.72) | 7.60 (6.06, 9.14) | 3.82 (2.92, 4.72) | 3.15 (2.41, 3.89) | 3.81 (3.00, 4.62) |
| **Sex** | | | | | | |
| Males | 6.45 (5.04, 7.86) | 5.20 (4.07, 6.34) | 6.09 (4.79, 7.38) | 4.02 (3.22, 4.81) | 3.43 (2.61, 4.25) | 3.32 (2.51, 4.14) |
| Females | 7.34 (5.74, 8.95) | 6.21 (4.93, 7.49) | 9.01 (7.11, 10.90) | 3.64 (2.41, 4.87) | 2.89 (1.87, 3.91) | 4.27 (3.10, 5.43) |
| **Age group** | | | | | | |
| 20-39 years | 0.35 (0.03, 0.67) | 0.49 (0.17, 0.81) | 0.42 (0.10, 0.75) | 1.85 (0.79, 2.91) | 2.03 (1.04, 3.03) | 1.67 (0.70, 2.65) |
| 40-64 years | 3.61 (2.30, 4.92) | 2.64 (1.60, 3.68) | 3.23 (2.06, 4.40) | 3.61 (2.30, 4.92) | 2.64 (1.60, 3.68) | 3.23 (2.06, 4.40) |
| ≥ 65 years | 26.12 (22.11, 30.13) | 22.03 (18.62, 25.44) | 30.26 (26.20, 34.32) | 7.85 (5.81, 9.90) | 6.29 (4.58, 8.00) | 8.97 (6.93, 11.01) |
| **Race** | | | | | | |
| White | 8.52 (6.66, 10.38) | 6.44 (5.06, 7.83) | 8.97 (7.04, 10.89) | 4.25 (2.93, 5.57) | 2.90 (1.86, 3.95) | 3.80 (2.60, 5.00) |
| Black | 6.80 (5.05, 8.55) | 9.69 (7.97, 11.41) | 11.32 (9.22, 13.42) | 4.96 (3.65, 6.27) | 7.61 (6.29, 8.92) | 8.19 (6.44, 9.95) |
| Multiple/Other | 3.15 (2.29, 4.01) | 2.42 (1.79, 3.05) | 2.86 (2.02, 3.70) | 2.33 (1.61, 3.05) | 1.93 (1.31, 2.55) | 2.07 (1.45, 2.69) |

no fluctuations. In contrast, the NHANES data showed a more modest increase in prevalence, accompanied by fluctuations. This finding aligned with previous research on neglected tropical diseases and is largely attributed to the systematic analytical framework [6]. Furthermore, the methodological framework of GBD, in which input data not reported by sex were adjusted using a specific ratio, amplified subgroup disparities. Similar patterns were observed in 16 other countries (S4 Table) [18–34], with notable differences in certain countries, such as Portugal [29]. Notably, many countries lack high-quality, population-based studies, forcing estimates to rely on data from higher-level geographical hierarchies. Caution is therefore necessary when interpreting the obtained results. Furthermore, stage-specific burden assessment is essential as it facilitates monitoring disease trends, optimizes healthcare resource allocation, and provides more accurate data for large-scale epidemiological studies.

It is also crucial to emphasize that estimates from GBD, NHANES, and other epidemiological studies, may overestimate CKD prevalence, as most source data lack repeated measurements of eGFR and albuminuria to confirm chronicity [8]. The ascertainment bias may arise when self-selected populations are included in studies, skewing prevalence reports for the general population [7]. For instance, individuals concerned about kidney disease or with a family history may voluntarily participate in screening programs. The limitations of using the GBD database to assess CKD burden have been thoroughly discussed in previous publications by the GBD Chronic Kidney Disease Collaboration [8]. However, recent studies have lacked sufficient descriptions of these limitations.

We also assessed the potential impact of definitions and equations on CKD burden estimates. The use of age-adjusted thresholds significantly reduced the prevalence of low eGFR, with more older women reclassified into non-CKD categories. Kidney function naturally declines with aging [35]. The incidence of eGFR<60 ml/min/1.73m² significantly increases among individuals aged 70 and older, with an incidence of 52.5% in those aged 80 and above [36]. Therefore, an eGFR<60 ml/min/1.73m² may represent a physiological condition in older adults. For older adults with only mild physiological GFR decline and no signs of increased urinary ACR or other renal damage, investigations, referrals, or therapeutic interventions with potential side effects are unnecessary and may even impose additional financial burdens. Moreover, the fixed GFR threshold may lead to missed CKD diagnoses in younger individuals whose renal function is compromised due to congenital conditions or adverse treatment histories. Age-adapted criteria represent an effective strategy to enhance the identification of pathology in younger adults with potential kidney damage while reducing overdiagnosis in older adults due to age-related GFR decline. The "normal" range of renal function is one consideration in CKD diagnosis, with another critical factor being the risk of future adverse outcomes [37]. Age-adapted criteria offer a practical approach to assessing

**Table 4. Number and weighted percentage of participants reclassified after applying age-adapted thresholds.**

| Equation | Characteristic | Number | Weighted percentage (%) |
|---|---|---|---|
| CKD-EPI$_{2009}$ | Age: 20–39 years<br>eGFR: 60–74 ml/min/1.73m$^2$ | 17 | 0.54 (0.20, 0.88) |
| | Males | 12 | 0.71 (0.28, 1.15) |
| | Females | 5 | 0.38 (0, 0.85) |
| | Age: ≥ 65 years<br>eGFR: 45–59 ml/min/1.73m$^2$ | 212 | 3.64 (2.91, 4.37) |
| | Males | 108 | 3.15 (2.38, 3.92) |
| | Females | 104 | 4.09 (3.12, 5.05) |
| CKD-EPI$_{2021}$ | Age: 20–39 years<br>eGFR: 60–74 ml/min/1.73m$^2$ | 19 | 0.56 (0.21, 0.91) |
| | Males | 15 | 0.79 (0.32, 1.27) |
| | Females | 4 | 0.34 (0, 0.80) |
| | Age: ≥ 65 years<br>eGFR: 45–59 ml/min/1.73m$^2$ | 198 | 3.13 (2.52, 3.75) |
| | Males | 103 | 2.57 (2.02, 3.12) |
| | Females | 95 | 3.66 (2.58, 4.74) |
| EKFC | Age: 20–39 years<br>eGFR: 60–74 ml/min/1.73m$^2$ | 15 | 0.45 (0.11, 0.79) |
| | Males | 11 | 0.57 (0.16, 0.98) |
| | Females | 4 | 0.34 (0, 0.80) |
| | Age: ≥ 65 years<br>eGFR: 45–59 ml/min/1.73m$^2$ | 253 | 4.24 (3.21, 5.27) |
| | Males | 126 | 3.34 (2.58, 4.10) |
| | Females | 127 | 5.07 (3.63, 6.52) |

this risk. Research by Ma et al. [38] has shown that age-adapted CKD criteria are more closely associated with cardiovascular risk factors and CKD-related comorbidities. Despite being controversial and challenging to implement, these criteria should be considered in future epidemiological research and clinical practice.

The KDIGO guidelines recommend avoiding the inclusion of race in eGFR calculations. CKD-EPI$_{2021}$ and EKFC have increased the estimated prevalence of low eGFR among Black adults, promoting a more equitable allocation of healthcare resources. The EKFC equation offers additional advantages, including relatively higher accuracy and significantly smaller bias [17]. This advantage partly stems from its foundational model, which assumes no age-dependent decline in kidney function before the age of 40, whereas CKD-EPI was developed based on the concept of a gradual decline in GFR starting at 18 years of age. Current research based on healthy participants suggests that GFR remains stable or declines slowly before the age of 40–50 due to adequate renal reserve [11,39]. Race-specific Q-values provide an additional option for individuals requiring more accurate estimates. Furthermore, modifications to the CKD-EPI only account for self-identified Black Americans geographically located in the United States [40]. In contrast, the EKFC equation is more flexible, as it can be applied to diverse populations provided Q-values are available. Existing validations have demonstrated its applicability [41,42]. In global epidemiological studies like GBD, the EKFC equation may be particularly suitable, enhancing inter-regional comparability. Further research is needed to calculate and validate population-specific Q-values.

The study has several limitations. First, the GBD study does not stratify CKD by stage. Additionally, the NHANES database lacks data on kidney transplants, and there is a significant amount of missing data related to dialysis (variable name: KIQ025), although this is unlikely to significantly affect the results. Finally, our comparisons were limited to databases

or studies that included nationally representative samples, and did not extend to regional-level data. Further research is needed to address these gaps in future studies.

## Conclusion

In conclusion, the GBD study provides a unique perspective on the global CKD burden, but we emphasize the importance of appropriate use and respect for local data rather than defaulting to GBD estimates without consideration. Standards and equations are critical components of CKD estimation, in which age-adapted thresholds should be considered, and the flexible EKFC equation holds promise for future applications in epidemiological research and clinical practice.

## Supporting information

**S1 Fig.  Flow diagram of participants' inclusion.**
(DOCX)

**S1 Table.  Equations to predict GFR.**
(DOCX)

**S2 Table.  Correction for serum creatinine in NHANES.**
(DOCX)

**S3 Table.  Annual estimated number and rate of CKD prevalence in US adults.**
(DOCX)

**S4 Table.  Reported results and corresponding GBD estimates of CKD prevalence across different countries.**
(DOCX)

**S1 File.  Literature review.**
(DOCX)

## Acknowledgments

We thank all doctors, epidemiologists, statisticians, or other related persons who devoted their time and energy to the establishment and accomplishment of the GBD study rounds.

## Author contributions

**Conceptualization:** Yao Ma.

**Formal analysis:** Yao Ma.

**Funding acquisition:** Weihong Zhao.

**Methodology:** Yao Ma, Xiang Wang.

**Project administration:** Weihong Zhao.

**Writing – original draft:** Yao Ma, Xiang Wang.

**Writing – review & editing:** Yao Ma, Weihong Zhao.

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
