## [Decision Letter · Decision Letter 0]

29 May 2025

Dear Dr. Zhao,

Thank you for submitting your manuscript to PLOS ONE. After careful consideration, we feel that it has merit but does not fully meet PLOS ONE’s publication criteria as it currently stands. Therefore, we invite you to submit a revised version of the manuscript that addresses the points raised during the review process.

We look forward to receiving your revised manuscript.

Kind regards,

Donovan Anthony McGrowder, PhD., MA., MSc

Academic Editor

PLOS ONE

Journal Requirements:

“This study was supported by National Key Research and Development Program of China (2023YFC3605500) and National Natural Science Foundation of China (82171585, 81971320).”

“This study was supported by National Key Research and Development Program of China (2023YFC3605500) and National Natural Science Foundation of China (82171585, 81971320).”

“This study was supported by National Key Research and Development Program of China (2023YFC3605500) and National Natural Science Foundation of China (82171585, 81971320).”

Additional Editor Comments (if provided):

Dear Dr. Zhao,

Thank you for submitting your manuscript entitled “Disparities in Chronic Kidney Disease Burden Estimates: From Different Sources, Definitions, and Equations” to PLOS ONE.

Your submission has been evaluated by five independent reviewers. They have provided detailed feedback and raised several important points that we believe, if addressed, will significantly enhance the quality and clarity of your manuscript. The reviewers’ comments are included below for your consideration.

We encourage you to submit a thoroughly revised version of your manuscript that fully addresses the reviewers' concerns. Please ensure that your resubmission is received within the timeframe indicated.

We appreciate your contribution to the field and look forward to receiving your revised manuscript.

Best regards,

Dr. Donovan McGrowder

Reviewers' comments:

Reviewer's Responses to Questions

**Comments to the Author**

1. Is the manuscript technically sound, and do the data support the conclusions?

Reviewer #1: Yes

Reviewer #2: Yes

Reviewer #3: Partly

Reviewer #4: Yes

Reviewer #5: Yes

2. Has the statistical analysis been performed appropriately and rigorously?

Reviewer #1: Yes

Reviewer #2: I Don't Know

Reviewer #3: I Don't Know

Reviewer #4: Yes

Reviewer #5: Yes

3. Have the authors made all data underlying the findings in their manuscript fully available?

Reviewer #1: Yes

Reviewer #2: Yes

Reviewer #3: No

Reviewer #4: Yes

Reviewer #5: Yes

4. Is the manuscript presented in an intelligible fashion and written in standard English?

Reviewer #1: Yes

Reviewer #2: Yes

Reviewer #3: No

Reviewer #4: Yes

Reviewer #5: Yes

Reviewer #1: It is an interesting article about the Disparities in chronic kidney disease burden estimates. I recommend below changes -

- correct definition of ckd - page 14- UACR is NOT always high in CKD and could be normal too

- Explain the rationale and importance ( for non nephrologist) behind age quartile egfr for 40 years old, 40-65 and > 65 .

Reviewer #2: This article aims to show the discrepancies in estimating renal function using different types of equations, and demographic details like sex, age and race. This is an important article as policy making is often based on epidemiological studies.

The methodology is complex as it looks at data over a 20 year period from different databases. It is unclear to me what actual numbers of data were extracted and what numbers were excluded due to insufficient data. Perhaps this can be clarified so that the statistics can be easier to understand. Overall, it is an important study that shows that epidemiological data must be critically analyzed before accepting the results as 'hard truths'

Reviewer #3: Ma Y et al described disparity of estimation on CKD by different source, definition and equation. They used data of GBD study 2021 and NHANES from 1999 to 2018 to analyze the prevalence of CKD in USA adults. They used formula of CKD-EPI 2009, CKD-EPI 2021 and EKFCRF. They calculated estimated annual percentage change of prevalence of CKD. They found difference across databases and estimating equations. These results are interesting but the following points are needed to be addressed.

1. In Table 2, the authors show the annual averages of the CKD prevalence. They should show the data in each year, as shown in Fig. 1.

2. In Table 3, the prevalence of CKD was dramatically decreased to 5.71 to 7.60% in fixed threshold (Definition of CKD <60 mL/min/1.73m2). As mentioned in Results section, the prevalence of CKD reduced by 50% compared with fixed threshold. In Table 2, however, the prevalence of CKD 2017-2018 was 14.1% to 15.6%. I do not understand this description.

3. The similar inconsistency was observed in the prevalence of CKD of NHANES 2017-2018 defined by age-adapted thresholds between Table 2 and Table 3.

4. Excellent performance of EKFC equation in US cohorts was extensively demonstrated by Delanaye P et al in Kidney International 2024 ( Ref 15). The authors could show the prevalence of CKD by EKFCPS. Q value was provide by NHANES, (Black men and women were 1.03 and 0.72mg/dl, respectively. The non-Black men and women were 0.99 and 0.71mg/dL, respectively).

Reviewer #4: The authors compared the epidemiological descriptions of CKD between different database, CKD definitions and eGFR equations. Although the findings were meaningful, there are several comments.

I completely agree with the idea that accurate estimates of the spatiotemporal patterns and trends in CKD prevalence are crucial for understanding the disease burden and improving CKD prevention and management. Additionally, they concluded that they emphasize the importance of appropriate use and respect for local data rather than defaulting to GBD estimates without consideration. Please propose an optimal method for accurately assessing the CKD burden.

Since there are huge results in the present study, I recommend showing the summary of results in the Discussion. Especially, please present briefly the findings that support the disparities of long-term trends between different databases in the Discussion.

Please explain database structure of GBD study including extracting methods of clinical data, management institution or type of participants. Do all participants in the GBD study have data on urine albumin, because CKD was defined as lower eGFR or higher ACR?

The terms "end-stage renal disease" and "end-stage kidney disease" are used inconsistently.

Reviewer #5: This manuscript presents a comprehensive analysis of the disparities in chronic kidney disease (CKD) prevalence estimates derived from different data sources (GBD vs. NHANES), GFR estimating equations (CKD-EPI 2009, CKD-EPI 2021, EKFC), and definitional thresholds (fixed vs. age-adapted). The topic is highly relevant in the context of increasing global reliance on epidemiological models for disease burden assessment. The manuscript is well-structured and methodologically sound. It highlights important implications for both public health surveillance and clinical nephrology, particularly regarding the potential for over- or underestimation of CKD prevalence depending on the choice of equation and threshold.

I recommend minor revision prior to acceptance. The authors are encouraged to strengthen the discussion of (1) clinical implications of reclassification, (2) relevance for aging East Asian populations, and (3) the need for stage-specific burden assessment in future work. This is a timely and valuable contribution to global nephrology and public health research.

**Do you want your identity to be public for this peer review?** For information about this choice, including consent withdrawal, please see our Privacy Policy

Reviewer #1: No

Reviewer #2: **Yes: ** Ngozi Virginia Aikpokpo

Reviewer #3: No

Reviewer #4: No

Reviewer #5: No

---

## [Author Response · Author response to Decision Letter 1]

25 Jun 2025

Dear Editor and Reviewers,

Thank you for your giving us an opportunity to revise our manuscript. On behalf of all the contributing authors, I would like to express our sincere appreciation of your letter and reviewers’ constructive comments concerning our manuscript entitled “Disparities in chronic kidney disease burden estimates: From different sources, definitions, and equations” (PONE-D-25-24676). These opinions help to improve the academic rigor of our article. We have studied these comments carefully and made modifications which we hope meet with your approval.

Revised portions are marked red in the manuscript. All comments are laid out below in italicized font and specific concerns have been numbered. Point-by-point responses to all nice reviewers are listed below in blue.

All authors have read and approved the re-submission of the manuscript. If you have any questions, please let me know.

Thank you for your consideration of our paper and we are looking forward to hearing from you.

Sincerely yours,

Weihong Zhao

zhaoweihongny@njmu.edu.cn

Response to Editor

We sincerely appreciate your consideration of our manuscript and deeply apologize for the negligence in format. All the issues have been addressed. We sincerely hope this revised manuscript will meet your requirements.

Response to Reviewer 1

Reviewer 1:

It is an interesting article about the Disparities in chronic kidney disease burden estimates. I recommend below changes.

Comment 1. Correct definition of ckd - page 14- UACR is NOT always high in CKD and could be normal too.

Response 1. Thank you very much for your comments. We have thoroughly reviewed the manuscript and confirmed that precise expressions such as 'and/or' and 'at least one' have been consistently used in relation to the definition of CKD.

Comment 2. Explain the rationale and importance (for non-nephrologist) behind age quartile egfr for 40 years old, 40-65 and > 65.

Response 2. Thank you very much for your professional advice. We sincerely apologize for the lack of a detailed description regarding age group stratification and the corresponding GFR thresholds in the manuscript. Dividing the population into younger adults, middle-aged adults, and older individuals using age cutoffs of 40 and 65 years is a commonly employed approach. However, there are more considerations in the field of nephrology.

1) In healthy individuals, GFR changes with age. The age of 40 is considered a pivotal point, as there is no age-dependent decline in renal function prior to this age. However, in older adults (65 years and older), a noticeable change in GFR occurs.

2) The expected values of GFR vary by age group. By setting age-adapted thresholds, we can more accurately identify the risk of adverse outcomes, particularly for younger individuals who may have poorer renal function due to congenital conditions and for older adults who experience physiologic age-related decline in GFR. Age groups and age-specific thresholds are derived from reference intervals of GFR (J Am Soc Nephrol. 2019;30(10):1785-1805).

Following your suggestion, we have added the relevant details to the Methods section. We sincerely hope that the revised version meets your requirements.

Response to Reviewer 2

Reviewer 2:

Comment 1. This article aims to show the discrepancies in estimating renal function using different types of equations, and demographic details like sex, age and race. This is an important article as policy making is often based on epidemiological studies.

The methodology is complex as it looks at data over a 20 year period from different databases. It is unclear to me what actual numbers of data were extracted and what numbers were excluded due to insufficient data. Perhaps this can be clarified so that the statistics can be easier to understand. Overall, it is an important study that shows that epidemiological data must be critically analyzed before accepting the results as 'hard truths'.

Response 1. Firstly, we would like to express our sincerest gratitude for your positive comments. Your feedback holds great significance for both our team and the readers. Then, we deeply apologize for the lack of detailed inclusion and exclusion processes in the manuscript.

The GBD study is the result of collaboration among researchers from around the world. The data used in the GBD study is extensive, sourced from international organizations (such as the WHO), governments, hospital networks, and published epidemiological data, though specific details have not been fully disclosed. The overall framework and search strategy for kidney and urinary diseases can be found in the published article (BMC Med Res Methodol. 2018;18(1):110). The methods employed in the GBD study are complex and difficult to describe comprehensively in this manuscript; thus, we have only provided a brief summary in the Methods section. These estimates can be directly accessed via the official website (https://vizhub.healthdata.org/gbd-results/), where the prevalence of CKD in target population can be retrieved by specifying parameters such as age, sex, and region. In the NHANES, we extracted demographic data, serum creatinine, urinary albumin, and urinary creatinine for all participants across the years 1999–2018 (covering 10 cycles). Individuals with missing data or those under the age of 20 were excluded from the analysis. A detailed flowchart outlining this process has been added in the Supplementary Material. We hope this revised manuscript will meet your requirements.

Response to Reviewer 3

Reviewer 3:

Ma Y et al described disparity of estimation on CKD by different source, definition and equation. They used data of GBD study 2021 and NHANES from 1999 to 2018 to analyze the prevalence of CKD in USA adults. They used formula of CKD-EPI 2009, CKD-EPI 2021 and EKFCRF. They calculated estimated annual percentage change of prevalence of CKD. They found difference across databases and estimating equations. These results are interesting, but the following points are needed to be addressed.

Comment 1. In Table 2, the authors show the annual averages of the CKD prevalence. They should show the data in each year, as shown in Fig. 1.

Response 1. Thank you very much for your comments. Table S3 presents the CKD cases and prevalence estimated using different databases, definitions, and equations in each year.

Comment 2. In Table 3, the prevalence of CKD was dramatically decreased to 5.71 to 7.60% in fixed threshold (Definition of CKD <60 mL/min/1.73m2). As mentioned in Results section, the prevalence of CKD reduced by 50% compared with fixed threshold. In Table 2, however, the prevalence of CKD 2017-2018 was 14.1% to 15.6%. I do not understand this description.

Response 2. Thank you very much for your question. We sincerely apologize for any confusion caused by unclear expressions. Throughout the manuscript, we have used two distinct concepts: ‘CKD’ and ‘low eGFR’. The former encompasses both eGFR and ACR, while the latter is defined as eGFR falling below the specified threshold(s).

In Table 2 and Table S3, we presented the impact of different databases, equations, and definitions on the estimated CKD cases and prevalence. The values were relatively larger. Since the choice of equation and definition primarily affects GFR, we further explored the changes in the prevalence of low eGFR (Table 3). The value of 50% was derived based on the data presented in Table 3.

Once again, thank you for your valuable comments. To avoid any further misunderstanding, we have included a clearer definition of low eGFR. We sincerely hope that the revised manuscript meets your requirements.

Comment 3. The similar inconsistency was observed in the prevalence of CKD of NHANES 2017-2018 defined by age-adapted thresholds between Table 2 and Table 3.

Response 3. We have provided the definition of low eGFR in the relevant section to avoid any potential misunderstandings. Once again, we sincerely appreciate your thorough review.

Comment 4. Excellent performance of EKFC equation in US cohorts was extensively demonstrated by Delanaye P et al in Kidney International 2024 (Ref 15). The authors could show the prevalence of CKD by EKFCPS. Q value was provided by NHANES, (Black men and women were 1.03 and 0.72mg/dl, respectively. The non-Black men and women were 0.99 and 0.71mg/dL, respectively).

Response 4. We greatly appreciate your professional suggestions and fully agree with your point that using race-specific Q values could yield more accurate eGFR estimates. We conducted preliminary calculations and found that the use of the EKFCPS equation did indeed lead to changes in CKD prevalence. However, we ultimately decided not to include this in the manuscript for two main reasons:

1) The race factor is a complex social construct distinct from biological variables such as ancestry. The KDIGO guidelines do not recommend incorporating race as a factor when assessing renal function (Table 1). The purpose of race-specific Q values is to provide more accurate GFR estimates for individuals seeking personalized assessments, rather than for large-scale evaluations. Therefore, using the EKFCPS equation in this study could potentially lead to misinterpretation, particularly for clinicians or researchers outside the nephrology field. It is worth noting that we consulted with Prof. Delanaye before commencing this study, and he also supported the use of the EKFCRF.

2) The primary objective of this study is to compare the impact of different equations on CKD prevalence estimates, rather than to assess their accuracy or precision. We are concerned that including too many details may obscure the main focus for the readers.

So, the EKFCPS equation was not included in this study, however, we have provided additional clarification in the Discussion section. We hope for your understanding.

Response to Reviewer 4

Reviewer 4:

The authors compared the epidemiological descriptions of CKD between different database, CKD definitions and eGFR equations. Although the findings were meaningful, there are several comments.

Comment 1. I completely agree with the idea that accurate estimates of the spatiotemporal patterns and trends in CKD prevalence are crucial for understanding the disease burden and improving CKD prevention and management. Additionally, they concluded that they emphasize the importance of appropriate use and respect for local data rather than defaulting to GBD estimates without consideration. Please propose an optimal method for accurately assessing the CKD burden.

Response 1. We would like to express our sincerest gratitude for your positive comments. Your feedback holds great significance for both our team and the readers.

The sources of data, the CKD definition, and the GFR estimating equations are critical factors in assessing the CKD burden, and they represent the three main aspects discussed in this study. For the broader general population or specific groups, we emphasize the importance of using and respecting local data, rather than defaulting to GBD estimates without due consideration. This is particularly crucial for countries lacking high-quality, population-based studies, where the priority should be to establish appropriate cohorts. Regarding CKD definition, we recommend using age-adapted thresholds, which can enhance the pathological identification of kidney damage in younger individuals, while minimizing overdiagnosis in older adults due to age-related declines in GFR. In terms of equation selection, we advocate for the flexible and accurate EKFC equation, which may be especially suitable in large-scale epidemiological studies, improving comparability across regions.

For diagnosing CKD in individuals, we similarly recommend the use of age-adapted thresholds and the EKFC equation. Additionally, race-specific Q-values provide an alternative option for individuals seeking more accurate estimates.

According to your comments, we have made additions to the Discussion section. We hope this revised manuscript will meet your requirements.

Comment 2. Since there are huge results in the present study, I recommend showing the summary of results in the Discussion. Especially, please present briefly the findings that support the disparities of long-term trends between different databases in the Discussion.

Response 2. Thank you very much for your professional comments. Given the broad scope of the topics covered in this study, and to assist readers in focusing on the key points, we have added a summary of the results.

Regarding long-term trends, the disparities primarily manifest in two aspects: changes in prevalence and fluctuations over time. As shown in Figure 1 and Table S3, the GBD study indicated a consistent annual increase in CKD prevalence, with no fluctuations observed. In contrast, the NHANES data showed a more modest increase in prevalence, accompanied by fluctuations. To assess the disparities in long-term trends, we calculated the corresponding EAPC, which reflects the trend of disease changes over a specific period. In the GBD database, both the EAPC and its 95%CI were greater than 0, indicating that the trend is statistically significant. Through a comprehensive literature review, we compared the GBD estimates with those reported in other studies and observed similar patterns. These findings align with previous research on neglected tropical diseases and are largely attributed to the systematic analytical framework. We have supplemented these findings in the Discussion section to help readers gain a better understanding.

Comment 3. Please explain database structure of GBD study including extracting methods of clinical data, management institution or type of participants. Do all participants in the GBD study have data on urine albumin, because CKD was defined as lower eGFR or higher ACR?

Response 3. Thank you very much for your comments. A unique perspective on the global burden of CKD is provided by the GBD collaboration, who have undertaken the monumental task of cataloguing the worldwide epidemiology and burden of communicable and non-communicable diseases since 1990. The data used in the GBD study is extensive, sourced from international organizations (such as the WHO), governments, hospital networks, and published epidemiological data, though specific details have not been fully disclosed. The overall framework and search strategy for kidney and urinary diseases can be found in the published article (BMC Med Res Methodol. 2018;18(1):110). The methods employed in the GBD study are complex and difficult to describe comprehensively in this manuscript; thus, we have only provided a brief summary in the Methods section. As for data usage, the official website (https://vizhub.healthdata.org/gbd-results/) allows users to directly retrieve and utilize CKD prevalence data for target populations by adjusting parameters such as age, sex, and region.

Regarding your other question, the GBD study 2021 defines CKD as individuals with an eGFR<60 ml/min/1.73m² or ACR >30 mg/g, including those with end-stage kidney disease who are on dialysis or have undergone a transplant (https://www.healthdata.org/research-analysis/diseases-injuries-risks/factsheets/2021-chronic-kidney-disease-level-3-disease). Studies that did not measure ACR were excluded from the GBD study (Lancet. 2020;395(10225):709-733). Detailed data sources are available for further review on GHDx (https://ghdx.healthdata.org).

Once again, we greatly appreciate your professional suggestions. We have made additional clarifications in the Methods section and hope that this revised manuscript will meet your requirements.

Comment 4. The terms "end-stage renal disease" and "end-stage kidney disease" are used inconsistently.

Response 4. We sincerely apologize for the oversight in the manuscript. We have standardized the terms to ‘end-stage renal disease’.

Response to Reviewer 5

Reviewer 5:

Comment 1. This manuscript presents a comprehensive analysis of the disparities in chronic kidney disease (CKD) prevalence estimates derived from dif

---

## [Editor Report · Decision Letter 1]

4 Jul 2025

Disparities in chronic kidney disease burden estimates: from different sources, definitions, and equations

PONE-D-25-24676R1

Dear Dr. Zhao,

We’re pleased to inform you that your manuscript has been judged scientifically suitable for publication and will be formally accepted for publication once it meets all outstanding technical requirements.

Kind regards,

Donovan Anthony McGrowder, PhD., MA., MSc

Academic Editor

PLOS ONE

Additional Editor Comments (optional):

Dear Dr. Zhao,

The manuscript entitled “Disparities in chronic kidney disease burden estimates: from different sources, definitions, and equations” was revised in accordance with the reviewers’ comments and is provisionally accepted pending final checks for formatting and technical requirements.

Regards,

Prof. Donovan McGrowder (Academic Editor)

---

## [Editor Report · Acceptance letter]

PONE-D-25-24676R1

PLOS ONE

Dear Dr. Zhao,

I'm pleased to inform you that your manuscript has been deemed suitable for publication in PLOS ONE. Congratulations! Your manuscript is now being handed over to our production team.

Kind regards,

on behalf of

Dr. Donovan Anthony McGrowder

Academic Editor

PLOS ONE